# Effects of the Introduction of Modern Immunotherapy on the Outcome of Isolated Limb Perfusion for Melanoma In-Transit Metastases

**DOI:** 10.3390/cancers15020472

**Published:** 2023-01-12

**Authors:** Carl-Jacob Holmberg, Jan Mattsson, Roger Olofsson Bagge

**Affiliations:** 1Department of Surgery, Sahlgrenska University Hospital, 413 45 Gothenburg, Sweden; 2Sahlgrenska Center for Cancer Research, Department of Surgery, Institute of Clinical Sciences, Sahlgrenska Academy, University of Gothenburg, 413 90 Gothenburg, Sweden; 3Wallenberg Centre for Molecular and Translational Medicine, University of Gothenburg, 405 30 Gothenburg, Sweden

**Keywords:** melanoma, ILP, immunotherapy

## Abstract

**Simple Summary:**

Melanoma in-transit metastasis has long been effectively treated with isolated limb perfusion (ILP), a procedure where the limb is treated with high concentrations of heated chemotherapy. The recent treatment revolution with the introduction of modern systemic treatments has changed how metastatic melanoma is treated. We analysed patients treated with ILP before the introduction of systemic immunotherapy (2010–2014) and compared them to those treated after (2017–2021). The patient population is largely unchanged, with only a slight increase in age. Importantly, there was no reduced effect of ILP, also in patients that previously had received and failed immunotherapy, showing that ILP is still a valid and important treatment for patients with melanoma in-transit metastasis.

**Abstract:**

Isolated limb perfusion (ILP) is an effective locoregional treatment for melanoma in-transit metastasis, but the advent of modern effective immunotherapy, such as ICI (immune checkpoint inhibitors), has changed the treatment landscape. The primary aims of this study were to compare the characteristics of the patient population receiving ILP before and after the introduction of modern systemic treatments and to assess if outcomes after ILP were influenced by previous immunotherapy treatment. A single-centre analysis of patients that underwent ILP for melanoma in-transit metastasis between 2010 and 2021 was conducted, with patients grouped and compared by treatment time period: pre-ICI era (2010–2014) and ICI era (2017–2021). 218 patients were included. Patients undergoing ILP in the ICI era were slightly older (median age 73 vs. 68 years) compared to the pre-ICI era, with no other difference found. The overall response rate (ORR) was 83% vs. 84% and the complete response (CR) rate was 52% vs. 47% for the pre-ICI era and the ICI era, respectively. For patients that had received and failed immunotherapy prior to ILP (n = 20), the ORR was 75% and the CR rate was 50%. Melanoma-specific survival has improved, with a 3-year survival rate of 54% in the pre-ICI era vs. 86% in the ICI era. The patient population undergoing ILP for in-transit melanoma is largely unchanged in the current era of effective systemic treatments. Response rates have not decreased, and prior ICI treatment did not affect response rates, making ILP still a valid treatment option for this patient group.

## 1. Introduction

Patients with melanoma have a risk of developing advanced locoregional metastasis, in-transit metastasis, that can be challenging to treat. Patients with low tumour burden can be treated with surgical excisions, but for patients with bulky or rapidly recurring diseases, either systemic or locoregional treatment options are instead recommended [1,2,3]. The introduction of effective systemic treatments, primarily ICI, and targeted therapies such as BRAF/MEK inhibitors, have opened new treatment avenues for unresectable melanoma [4,5,6,7]. As previously reported, both the efficacy of ICI on exclusively in-transit disease, as well as the optimal relation between ICI and the varied landscape of locoregional treatments available, are still largely unknown [8,9,10,11,12,13].

One locoregional treatment is ILP, which was pioneered in the 1950s and involves isolating an affected limb by tourniquet and perfusing it with heated chemotherapy at concentrations that would not be tolerated systemically [14,15]. Treatment results are robust, with reported ORR of 65–100% and CR rates of up to 65% [16,17,18,19,20]. The procedure can be repeated with new disease recurrences, adding further utility.

With the changing treatment landscape of melanoma with the introduction of effective systemic treatments, the historically reported outcomes of ILP and other loco-regional treatments can be questioned since there very likely has been a shift in patient selection. Hypothetically, patients would not be referred any longer for ILP as a first-line of treatment, but rather after failing systemic treatments. Also, the introduction of adjuvant treatments, both BRAF/MEK and ICI, could potentially change the patient population. Therefore, the primary aim of this study was to compare the characteristics of the patient population receiving ILP before and after the introduction of modern effective systemic treatments. A secondary aim was to specifically analyse those patients that had previously failed ICI treatment as a first-line treatment, and assess response, time to progression, toxicity, complications and survival after ILP.

## 2. Methods

A single centre analysis of a prospectively kept database of patients treated with first-time ILP for melanoma in-transit metastasis between 2010 and 2021 at our institution, the national referral centre for ILP in Sweden. Data on patient and tumour characteristics were collected, as well as response, toxicity, recurrence and survival. Patients were grouped in three different treatment periods: the pre-immunotherapy era (pre-ICI era, 2010–2014), the transition period (2015–2016) and the current immunotherapy era (ICI era, 2017–2021). The transition period was chosen as a “wash-out” period when modern effective systemic treatments were either introduced or used within clinical trials, but not used in routine healthcare. Since 2017, both BRAF/MEK and immunotherapy have been considered standard of care in Sweden, both CTLA-4 and PD-1 in monotherapy, as well as a combination treatment with both CTLA-4 and PD-1 inhibitors. In the pre-ICI era, ILP was considered the first-line treatment for ITM of the extremities, and patients did not receive any other treatment before ILP except surgical excisions. Statistical comparisons will exclude the transition period, and only compare the eras before and after the introduction of immunotherapy (pre-ICI vs. ICI era). Data on treatment before ILP is available, and therefore we could identify patients that had received treatment before ILP. Unfortunately, data concerning treatment after ILP is not routinely recorded, since patients are referred for ILP from all over Sweden, and we, therefore, choose to compare treatment eras.

The response was evaluated as the best response during follow-up according to the RECIST criteria modified for cutaneous lesions (allowing for calliper measurement if lesions were not visible on radiology) [21]. To be considered a complete response (CR), all lesions had to disappear. Partial response (PR) was defined as a decrease of more than 30% of the total tumour burden, measured as the number of lesions or shrinkage in the largest tumour diameter. Progressive disease (PD) was defined as an increase of more than 25% in existing lesions or the appearance of new lesions. Stable disease (SD) was defined as when criteria for CR, PR or PD were not met. Toxicity reactions after ILP were reported according to the Wieberdink scale and surgical complications according to the Clavien-Dindo classification [22,23].

Local progression was defined as the recurrence or progression of in-transit metastasis within the treated limb, and time to local progression was calculated from ILP to recurrence. Systemic progression was defined as systemic recurrences (M1 disease), and time to systemic progression was analysed only in the cohort of patients not already having M1 disease at the time of ILP. Survival was calculated from ILP to death or end of follow-up. Follow-up was performed according to the national Swedish guidelines, and the guidelines have not changed between the study periods. Time to recurrence and survival was estimated using the Kaplan-Meier method and the log-rank test. Mann-Whitney test was used for continuous non-parametric variables and Fisher’s Exact test was used for categorical variables. Statistical significance was set at *p* < 0.05. Statistical analysis was conducted using SPSS v28 (SPSS Inc., Chicago, IL, USA).

The study was conducted in adherence to the ethical principles of the Declaration of Helsinki and with the approval of the Swedish Ethical Review Authority (dnr 721–08).

## 3. Results

A total of 218 patients were treated with first-time ILP for melanoma in-transit metastases between 2010 and 2021. The median age was 71 years (IQR 56–86) with 51% (n = 112) females and 49% (n = 106) males (Table 1). When stratified by treatment time period, 96 patients received ILP during the time defined as the pre-ICI era, 37 patients during the transition period and 85 patients during the current ICI era. In the pre-ICI era, no patients received ICI before ILP, while in the ICI era, there were 20 patients (24%) that received ICI prior to ILP.

When analysing and comparing potential shifts in patient selection between the eras, the median age increased from 68 years in the pre-ICI era to 73 years in the ICI era (*p* = 0.03). There was no statistically significant change in the median number of metastases (5.0 vs. 7.5, *p* = 0.13), the median size of the largest metastasis (12.5 vs. 15.0 mm, *p* = 0.71), simultaneous lymph node dissections (31% vs. 19%, *p* = 0.06) or the number of patients with general metastatic disease (stage IV) at the time of ILP treatment (85% vs. 92%, *p* = 0.25). The time from diagnosis of primary melanoma to ILP was 28.0 months in the pre-ICI era and 27.0 months in the ICI era, also without a statistically significant difference.

### 3.1. Response Rates

The response was evaluable in 209 (96%) of all patients, with an ORR of 84% including 52% CR. When analysing responses based on era, the ORR was 83% vs. 84% (*p* = 1.00) and the CR rate was 52% vs. 47% (*p* = 0.54) for the pre-ICI era and the ICI era, respectively (Table 2). When specifically comparing the 20 patients that received immunotherapy before ILP to those who did not, the ORR was 75% vs. 85% (*p* = 0.52) and the CR rate was 50% vs. 51% (*p* = 1.00), respectively.

### 3.2. Local Progression

The median time to local progression for the whole study cohort was 29 months, with a 1- and 5-year local progression-free rate of 65% and 48% respectively. When stratified by treatment era, the median time to local progression was 10 months for the pre-ICI era and not yet reached for the ICI era group (*p* < 0.001) (Figure 1a). When specifically analysing the 20 patients that had previously received immunotherapy, the median time to local progression was not yet reached and the 1-year local progression-free survival was 100% (*p* = 0.13) (Figure 1b).

### 3.3. Systemic Progression

The median time to systemic progression for the whole study cohort was 71 months, with 1- and 5-year systemic progression rates of 72% and 57% respectively. When stratified by treatment era, the median time to systemic progression was 34 months for the pre-ICI era and not yet reached for the ICI era (*p* = 0.002) (Figure 1c). When specifically analysing the 20 patients that had previously received immunotherapy, the median time to systemic progression was not yet reached and the 1-year systemic progression-free survival rate was 95% (*p* = 0.53).

### 3.4. Melanoma-Specific Survival

Median MSS for the whole cohort of all included patients was not yet reached, with 1-, 5- and 10-year MSS rates of 81%, 58% and 51% respectively. The median MSS was 46 months for the pre-ICI era and not yet reached for the ICI era (*p* < 0.001) (Figure 2a), with 1-year MSS rates of 73% vs. 92% and 3-year MSS rates of 54% vs. 86% respectively. When comparing the patients in the ICI era that received immunotherapy or not, there was no difference in MSS (*p* = 0.51) (Figure 2b).

### 3.5. Toxicity and Complications

For the overall cohort, 92% (n = 201) of patients undergoing ILP experienced toxicity reactions Wieberdink grade I–III (mild or no reaction) (Table 3). Grade I–III toxicity was present in 91% (n = 87) and 93% (n = 79) of the patients in the pre-ICI era and ICI era, respectively (*p* = 0.27). In the overall cohort, 3% (n = 8) had Clavien-Dindo grade III–IV surgical complications (life-threatening or requiring surgical intervention) to the ILP procedure, with 2% (n = 2) vs. 4% (n = 3) grade III and 1% (n = 1) vs. 1% (n = 1) grade IV for pre-ICI and ICI era respectively.

## 4. Discussion

We conclude that after the introduction of modern immunotherapy, no major significant differences in referral patterns have emerged, except that there has been an increase in the median age of 5 years. However, the study groups are relatively small, increasing the risk of not identifying a true difference due to low statistical power. When analysing the trends there is notably a non-significant decrease in simultaneous lymph node dissections and patients with M1 disease, while the number of tumours and tumour size has increased. Taken together, there might have been a smaller shift in referral patterns, with more patients having isolated ITMs, that now also are referred at a relatively later stage with higher age and larger tumour burden. Secondly, and of high importance, is that there was no reduced effect of ILP when comparing the two eras, which was true also for patients that previously had received and failed immunotherapy, showing that ILP is still a valid and important treatment option for patients with melanoma in-transit metastasis.

We report an ORR of 84% and a CR rate of 52%, which is comparable to previously established response rates for ILP [16,17,18,19,20]. No statistically significant difference between the pre-ICI and ICI eras could be seen concerning response, time to local progression or time to systemic progression. In addition, there was no difference in toxicity or complications. There was, however, a difference in MSS where patients in the ICI era had significantly better disease-specific survival. This is an expected result and reflects the treatment revolution of the new effective systemic treatments for patients with melanoma.

When analysing the subgroup of patients that had received ICI prior to ILP, we found no significant difference in response, time to local progression, time to systemic progression, MSS or toxicity, compared to those that had not received prior ICI. As such, we conclude that failed previous ICI treatment does not appear to influence the outcome of ILP. Since it has been suggested, by us and others, that immunological factors seem to be of importance for treatment outcomes, the hypothesis was that a selection towards patients failing immunotherapy could lead to reduced response rates after ILP [24,25,26,27,28]. The finding that this was not the case, may suggest that different immunological pathways are involved.

Davies et al. have previously published a retrospective analysis of 97 patients that had undergone ILP, out of which 16 had received prior immunotherapy [29]. Notably, the authors included both ICI and oncolytic virotherapy using talimogene laherparepvec (TVEC) under the heading immunotherapy, and thus also included a small number of patients that had received TVEC only. Patient characteristics remained largely unchanged over time in this cohort, and the authors could find no indication that the patient selection for ILP had changed with the introduction of immunotherapy. However, in contrast to our current findings, they showed a significantly decreased CR rate, overall survival rate and distant progression-free survival rate in patients that had received immunotherapy prior to ILP. A possible explanation for these contrasting findings is a difference between institutions, where the number of referrals for ILP in Sweden has largely remained unchanged over the years, but where referrals in other countries have decreased significantly.

Ariyan et al. have reported appealing data supporting the combination of regional therapy and immunotherapy [30]. In this phase II trial, the authors attempted to translate into a clinical setting the preclinical findings that immunotherapy and chemotherapy in combination have an increased local pro-inflammatory effect at the tumour site. Twenty-six patients with advanced melanoma underwent isolated limb infusion (ILI), a treatment similar to ILP, and were then given systemic treatment with CTLA4 inhibition. The results showed an ORR of 85% at 3 months and 58% progression-free survival at 12 months, as well as increased tumour infiltration of T-cells. Similarly, a phase I study of TVEC together with the PD-1 inhibitor pembrolizumab showed a synergistic effect, and though a following phase III trial did show an increased response rate (18% vs. 12%), there was, however, no benefit in progression-free survival or overall survival [31,32]. Further trials are needed to evaluate the optimal combination and sequence of these different treatment modalities. Two examples of ongoing studies are the Nivo-ILP (ClinicalTrials.gov: NCT03685890) and NIVEC (ClinicalTrials.gov: NCT04330430) studies, which are recruiting patients with in-transit metastasis, examining the combination of ILP and nivolumab, and T-VEC and nivolumab, respectively.

A strength of this study is that it can be considered population-based since all patients in Sweden are referred to our centre only. This results in what we believe to be an accurate overview of the changes in referral patterns, but also similar evaluations over time concerning outcomes. As has been noted previously, measurement of treatment response in patients with in-transit metastasis is difficult. Many institutions report using RECIST criteria modified for cutaneous lesions, but do not specify the modifications further, and future collaborative work should focus on establishing standards specifically for in-transit metastases. The main limitations of the current study are the retrospective design, even though the data is collected prospectively, and the relatively low number of patients. Secondly, the external validity of the current results may be limited by the relatively high use of ILP in Sweden, where patients in other countries may follow other treatment plans and scheduling.

## 5. Conclusions

ILP remains an effective locoregional treatment option in the era of effective systemic treatments, where patients previously failing immunotherapy have similarly high response rates as treatment naïve patients. Further studies are needed to establish the optimal combination and timing of the high local response rates of locoregional treatments with the systemic effects of immunotherapy.

## Figures and Tables

**Figure 1 cancers-15-00472-f001:**
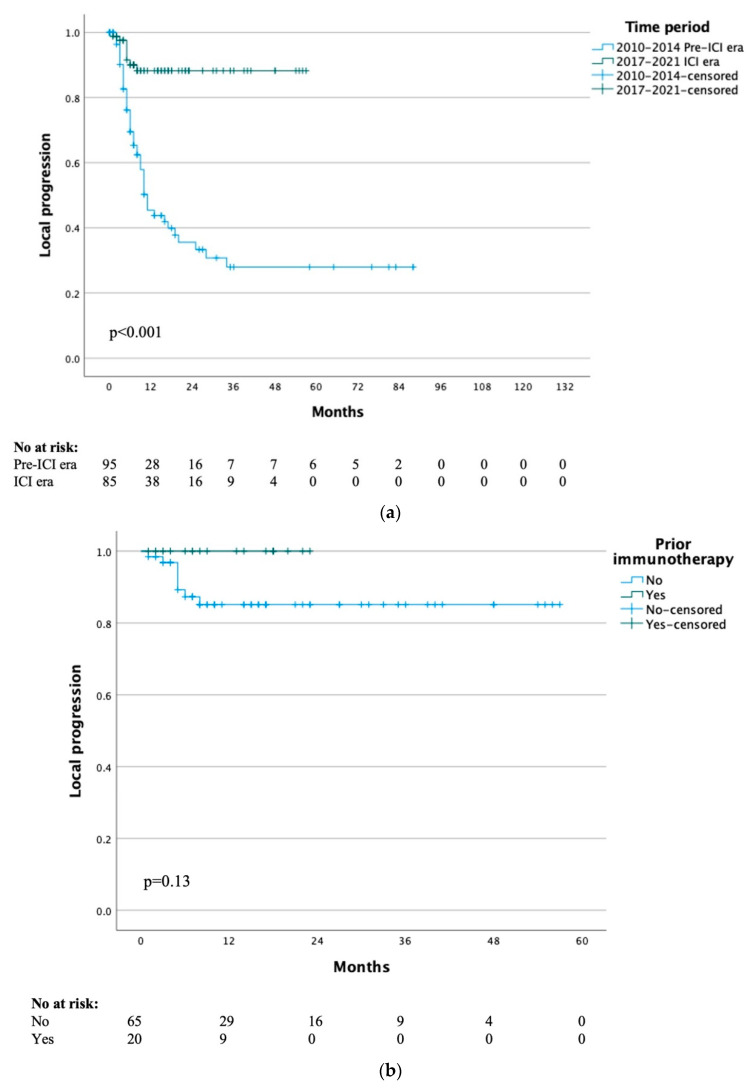
(**a**) Time to local progression by treatment time period. (**b**) Time to local progression of ICI era group by previous immunotherapy. (**c**) Time to systemic progression by treatment time period.

**Figure 2 cancers-15-00472-f002:**
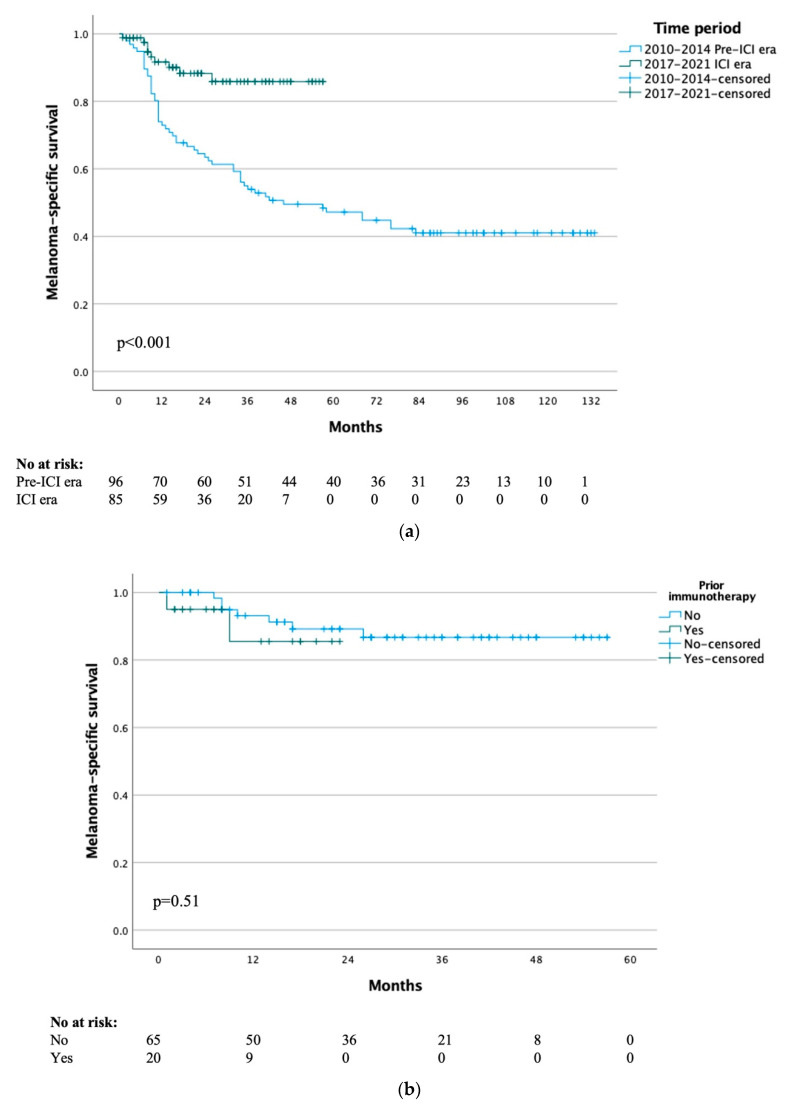
(**a**) Melanoma-specific survival by treatment time period. (**b**) Melanoma-specific survival of ICI era group by previous immunotherapy.

**Table 1 cancers-15-00472-t001:** Patient Characteristics.

Patient Variable	Pre-ICI Era (n = 96)	Transition Era (n = 37)	ICI Era (n = 85)	Overall (n = 218)	*p*-Value
**Age**, median (IQR)	68 years (60–78)	70 years (62–77)	73 years (67–79)	71 years (56–86)	*p* = 0.03
**Gender**, n (%)					
Male	47 (49%)	17 (46%)	42 (49%)	106 (49%)	*p* = 1.00
Female	49 (51%)	20 (54%)	43 (51%)	112 (51%)	
**Previous treatment with ICI,** n (%)	0 (0%)	4 (11%)	20 (24%)	24 (11%)	*p* < 0.01
**BRAF V600E/K mutated,** n (%)	19 (30%)	12 (35%)	24 (33%)	55 (25%)	*p* = 0.85
**Simultaneous lymph node dissection,** n (%)					*p* = 0.06
Yes	30 (31%)	8 (22%)	16 (19%)	54 (25%)	
No	66 (69%)	29 (78%)	69 (81%)	164 (75%)	
**M stage at ILP**, n (%)					*p* = 0.25
M0	82 (85%)	34 (92%)	78 (92%)	194 (89%)	
M1	14 (15%)	3 (8%)	7 (8%)	24 (11%)	
**Time from primary tumour to ILP**, median (range)	28.0 months (0–170)	27.5 months (4–102)	27.0 months (1–287)	27.0 months (0–287)	*p* = 0.26
**Number of metastases at time of ILP**, median (range)	5.0 (1–100)	6.0 (1–99)	7.5 (1–199)	6.0 (1–199)	*p* = 0.13
**Size of largest metastasis at time of ILP**, median (range)	12.5 mm (1–150)	10.0 mm (3–97)	15.0 mm (2–120)	14.0 mm (1–149)	*p* = 0.71
**Extremity treated with ILP**, n (%)					*p* = 0.69
Upper limb	17 (18%)	4 (11%)	13 (15%)	34 (16%)	
Lower limb	79 (82%)	33 (89%)	72 (85%)	184 (84%)	

**Table 2 cancers-15-00472-t002:** Response to ILP by Treatment Era (Evaluable Patients Only).

Best Response	Pre-ICI Era n (%)	Transition Era n (%)	ICI Era n (%)	Overall n (%)
**CR**	47 (52%)	22 (61%)	39 (47%)	108 (52%)
**PR**	28 (31%)	8 (22%)	31 (37%)	67 (32%)
**SD**	5 (6%)	3 (8%)	6 (7%)	14 (7%)
**PD**	10 (11%)	3 (8%)	7 (8%)	20 (10%)
**ORR**	75 (83%)	30 (83%)	70 (84%)	175 (84%)
**Missing**	6 (6%)	1 (3%)	2 (2%)	9 (4%)

**Table 3 cancers-15-00472-t003:** Toxicity and Complications to ILP by Treatment Era.

Wieberdink	Pre-ICI Era n (%)	Transition Era n (%)	ICI Era n (%)	Overall n (%)
I	4 (4%)	0 (0%)	1 (1%)	5 (2%)
II	52 (54%)	22 (60%)	46 (54%)	120 (55%)
III	31 (32%)	13 (35%)	32 (38%)	76 (35%)
IV	2 (2%)	1 (3%)	5 (6%)	8 (4%)
V	0 (0%)	1 (3%)	0 (0%)	1 (1%)
Missing	7 (7%)	0 (0%)	1 (1%)	8 (4%)
**Clavien-Dindo**				
0	86 (90%)	34 (92%)	72 (85%)	192 (88%)
I	2 (2%)	2 (5%)	6 (7%)	10 (5%)
II	5 (5%)	0 (0%)	3 (4%)	8 (4%)
III	2 (2%)	0 (0%)	3 (4%)	5 (2%)
IV	1 (1%)	1 (3%)	1 (1%)	3 (1%)

## Data Availability

Data is to be made available upon request.

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
