# Peer review of "Effects of the Introduction of Modern Immunotherapy on the Outcome of Isolated Limb Perfusion for Melanoma In-Transit Metastases"

_cancers, 2023, doi:10.3390/cancers15020472_

Round 1

Reviewer 1 Report (Previous Reviewer 4)

First of all, my congratulations to the authors. I believe that there is a clear attempt to improve, which has resulted in a more complete manuscript that leaves few doubts. The only drawback I see is that it is not possible to update the type of immunotherapy in the database. I think that it would provide a better perspective of the current situation and could help to specify relevant actions in different areas.

Reviewer 2 Report (Previous Reviewer 3)

Holmberg and Colleagues well analyzed the clinical practice and outcomes of ILP in high volume referral center. 

I find this manuscript could add relevant scientific information  

This manuscript is a resubmission of an earlier submission. The following is a list of the peer review reports and author responses from that submission.

Round 1

Reviewer 1 Report

Article: Effects of the introduction of modern immunotherapy on the outcome of isolated limb perfusion for melanoma in-transit metastases.

The aims of this study was to compare the characteristics of the patient population receiving isolated limb perfusion (ILP) before and after the introduction of modern effective systemic treatments, and to assess if outcomes after ILP, measured as response, time to progression, toxicity, complications and survival, was influenced in patients that had previously failed immune-checkpoint inhibitors (ICI) treatment. They concluded that ILP is still a valid and important treatment option for patients with melanoma in-transit metastasis since no major differences in referral patterns have emerged after the introduction of modern immunotherapy

The introduction, Results, Tables, figures, discussion, concluding remarks and references are framed correctly. Overall, the article is informative for the scientific fraternity. I recommend the paper for publication after rectifying the below mentioned minor errors

Line No. 92 :   for continues non-parametric

Line No. 104 : Patients that had

Line No.115:   patients (96%) of all patients,

Line No. 118   patients that had

Line No. 181   might point towards that different immunological pathways are involved

Reference:

In the reference No.26, Volume and page No. details not given

Author Response

Reviewer 1: The aims of this study was to compare the characteristics of the patient population receiving isolated limb perfusion (ILP) before and after the introduction of modern effective systemic treatments, and to assess if outcomes after ILP, measured as response, time to progression, toxicity, complications and survival, was influenced in patients that had previously failed immune-checkpoint inhibitors (ICI) treatment. They concluded that ILP is still a valid and important treatment option for patients with melanoma in-transit metastasis since no major differences in referral patterns have emerged after the introduction of modern immunotherapy 

The introduction, Results, Tables, figures, discussion, concluding remarks and references are framed correctly. Overall, the article is informative for the scientific fraternity. I recommend the paper for publication after rectifying the below mentioned minor errors 

Line No. 92 :   for continues non-parametric

Line No. 104 : Patients that had

Line No.115:   patients (96%) of all patients,

Line No. 118   patients that had

Line No. 181   might point towards that different immunological pathways are involved

Reference: In the reference No.26, Volume and page No. details not given

Response: Thank you for your well formulated feedback! All six minor remarks have been corrected.

Reviewer 2 Report

Effects of the introduction of modern immunotherapy on the outcome of isolated limb perfusion for melanoma in-transit metastases

Holmberg et al. collected and retrospectively analyzed clinical data of 208 patients with melanoma in-transit metastasis after treatment/s with isolated limb perfusion (ILP). The clinical data were collected from a single center, Sahlgrenska University Hospital in Gothenburg, Sweden.

The primary aims of this study were

1. To compare the characteristics of the patient population receiving ILP treatment/s before (pre-ICI era :2010-2014) and after (ICI era: 2017-2021) the introduction of modern systemic treatments.

2. To assess if outcomes after ILP were influenced by previous immunotherapy treatment.

·        The main drawback of the paper, which limits the ability to evaluate it, is the critical lack of details in the methods section. It is not clear what the immunotherapeutic treatment regimen received by the 65-patient cohort of the ICI era, is and what differentiates them from the 20 patients’ sub-group which received failed immunotherapy prior to the ILP treatments.

a.      Did they receive successful immunotherapy prior to the ILP treatment/s, in contrast to failed immunotherapy of the 20-patient’s subgroup? (i.e., the combination of a regional therapy and systemic immunotherapy). In this case, it is not clear why the ILP treatments were necessary for this cohort.

b.      Did they receive the immunotherapy treatments only after the ILP treatment/s? in contrast to the 20 patients’ sub-group which received the immunotherapy treatment prior to ILP treatment/s? In this case, what was the PFS for the ILP treatments for this subgroup of the ICI era?

c.      Did they receive immunotherapy at all? Or maybe for some of them, the local ILP treatments were enough for recovery? As suggested in line 58- “Hypothetically, patients would not be referred any longer for ILP as a first-line of treatment, but rather after failing systemic treatments”. If so, what is the point in comparing the success of ILP treatments between two groups which never received immunotherapy (65 out of 85 of the ICI-era did not receive immunotherapy)?

·        The authors declare at the end of the discussion section that in Sweden patients may follow other treatment plans and scheduling. Therefore, it is not clear what kind of immunotherapy treatments the ICI era cohort received: mono treatments? combination treatments? CTLA4-inhibitors? PD-1 inhibitors? and what other than ILP treatments did the pre-ICI era cohort receive? In addition, are the ILP treatments the first line treatments for melanoma in-transit metastasis in the ICI era in Sweden (“standard of care” -line 74)? If so, why do only 20 patients receive it as first-line treatment? Is it first-line treatment only for M0 patients?

·        “Systemic progression was defined as systemic recurrences (M1 disease), and time to systemic progression was analyzed only in the cohort of patients not already having M1 disease at the time of ILP” (lines 88-89). How many patients already had M1 disease? Since figure 1c legend lacks of details, this data cannot be extracted even from table 1 as we don’t know whether table 1 referred to the first ILP, the second ILP, or the last ILP. (“The procedure can be repeated with new disease recurrences, adding further utility”-line53).

·        One of the two primary aims of the research is to compare the characteristics of the patient population receiving ILP treatment/s before (pre-ICI era:2010-2014) and after (ICI era: 2017-2021) the introduction of modern systemic treatments. There is no explanation for why it is interesting or important. If indeed it is a primary aim of the research, why is there no discussion in the text of the 5 years increase in patients' median age between the pre-ICI era and the ICI era? Is that because ILP treatments are not first-line treatments in Sweden during the ICI era? Does this strengthen the conclusions?

·        It is not clear how from 20 patients who didn’t reach the median time to local progression (of which 15 are censored; Figure 1B), the authors conclude: “Importantly, there was no reduced effect of ILP, which was true also for patients that previously had received and failed immunotherapy, showing that ILP is still a valid and important treatment option for patients with melanoma in-transit metastasis.” (Line 164-166).

To conclude whether the pre-treatments have a positive or negative effect, the authors should use a larger cohort with longer follow-up times.

·        What is the point of the 10 years' survival statistics for all the patients (line 144) when most of the patients received the treatments less than 10 years ago? What is the point of the MSS statistical test between the pre-ICI era and the ICI era (p<0.05, line 145) when some unknown fraction of one of the groups receive the treatment 12 or 24 months before the statistical analysis?

·        Line 156, p<0.05 - It is not clear how the differences between 93% and 91% showed statistical significance?   

Additional minor comments:

1.      Table 1: At the PDF version of the paper mismatched parenthesis.

2.      Lack of title and details on figure legend e.g., p-value, n, etc.

3.      The abbreviation ILP is defined 3 times: lines 13, 21, 49.

4.      The abbreviation ORR is defined twice: lines 30, 52.

5.      The abbreviation CR is defined twice: lines 30, 53.

6.      The abbreviation ICI is first defined at line 44, but mentioned earlier on the text e.g., at line 35.

7.      The abbreviations PFS and OS at line 206 are not defined at the text at all.

8.      Line 115-116 and line 167: ORR= 84% CR=52% while in the table ORR=80% and CR=50% for all patients.

9.      At line 120- “respectively” should be added.

10.   “Number at risk” tables should be added to the Kaplan-Meier graphs. 

Author Response

Reviewer 2: Holmberg et al. collected and retrospectively analyzed clinical data of 208 patients with melanoma in-transit metastasis after treatment/s with isolated limb perfusion (ILP). The clinical data were collected from a single center, Sahlgrenska University Hospital in Gothenburg, Sweden.

The primary aims of this study were: 1. To compare the characteristics of the patient population receiving ILP treatment/s before (pre-ICI era :2010-2014) and after (ICI era: 2017-2021) the introduction of modern systemic treatments. 2. To assess if outcomes after ILP were influenced by previous immunotherapy treatment.

The main drawback of the paper, which limits the ability to evaluate it, is the critical lack of details in the methods section. It is not clear what the immunotherapeutic treatment regimen received by the 65-patient cohort of the ICI era is and what differentiates them from the 20 patients’ sub-group which received failed immunotherapy prior to the ILP treatments.

  1. Did they receive successful immunotherapy prior to the ILP treatment/s, in contrast to failed immunotherapy of the 20-patient’s subgroup? (i.e., the combination of a regional therapy and systemic immunotherapy). In this case, it is not clear why the ILP treatments were necessary for this cohort.

Response: We agree that this might have not been clear in our manuscript, and we have now tried to clarify this further in the last paragraph of the Introduction. The first aim was to evaluate any potential change in patient characteristics after the introduction of effective systemic treatments, where one hypothesis has been that patient referred for ILP has e.g. more advanced disease since they more often have received treatment before being referred, including adjuvant treatment. Here we have not specified the treatment received for each patient, but rather compared two distinct treatment eras, before and after the introduction of effective systemic treatment. The secondary aim was to specifically analyze those patients that had failed ICI, and then referred for ILP. There is e.g., one manuscript showing that these patients do worse, than those patients that are treatment naïve.

  1. Did they receive the immunotherapy treatments only after the ILP treatment/s? in contrast to the 20 patients’ sub-group which received the immunotherapy treatment prior to ILP treatment/s? In this case, what was the PFS for the ILP treatments for this subgroup of the ICI era?

Response: How patients have been treated after ILP is not recorded, and that is why we have chosen to compare treatment eras instead. We have now clarified this in the first paragraph of the Methods section.

  1. Did they receive immunotherapy at all? Or maybe for some of them, the local ILP treatments were enough for recovery? As suggested in line 58- “Hypothetically, patients would not be referred any longer for ILP as a first-line of treatment, but rather after failing systemic treatments”.If so, what is the point in comparing the success of ILP treatments between two groups which never received immunotherapy (65 out of 85 of the ICI-era did not receive immunotherapy)?

Response: The question we try to answer in this cohort is if patients that already had received immunotherapy, and failed, has worse outcomes than patients undergoing ILP as a first-line of treatment.

  1. The authors declare at the end of the discussion section that in Sweden patients may follow other treatment plans and scheduling. Therefore, it is not clear what kind of immunotherapy treatments the ICI era cohort received: mono treatments? combination treatments? CTLA4-inhibitors? PD-1 inhibitors? and what other than ILP treatments did the pre-ICI era cohort receive? In addition, are the ILP treatments the first line treatments for melanoma in-transit metastasis in the ICI era in Sweden (“standard of care” -line 74)? If so, why do only 20 patients receive it as first-line treatment? Is it first-line treatment only for M0 patients?

Response: Thanks for this question, we have now added this information in the first paragraph of the Methods section to make this more clear. 

  1. “Systemic progression was defined as systemic recurrences (M1 disease), and time to systemic progression was analyzed only in the cohort of patients not already having M1 disease at the time of ILP” (lines 88-89)How many patients already had M1 disease? Since figure 1c legend lacks of details, this data cannot be extracted even from table 1 as we don’t know whether table 1 referred to the first ILP, the second ILP, or the last ILP. (“The procedure can be repeated with new disease recurrences, adding further utility”-line53).

Response: Thanks for the question. As stated in the first line in the Methods section, only first-time ILP patients were included. As such, “M stage at ILP” in Table 1 denotes number of patients already in stage M1, overall 11% of the patients.

  1. One of the two primary aims of the research is to compare the characteristics of the patient population receiving ILP treatment/s before (pre-ICI era:2010-2014) and after (ICI era: 2017-2021) the introduction of modern systemic treatments. There is no explanation for why it is interesting or important. If indeed it is a primary aim of the research, why is there no discussion in the text of the 5 years increase in patients' median age between the pre-ICI era and the ICI era? Is that because ILP treatments are not first-line treatments in Sweden during the ICI era? Does this strengthen the conclusions?

Response: We agree that this is missing. We have now added this in the first paragraph of the Discussion, and we have also written out the non-significant differences in the Results section.

  1. It is not clear how from 20 patients who didn’t reach the median time to local progression (of which 15 are censored; Figure 1B), the authors conclude: “Importantly, there was no reduced effect of ILP, which was true also for patients that previously had received and failed immunotherapy, showing that ILP is still a valid and important treatment option for patients with melanoma in-transit metastasis.” (Line 164-166). To conclude whether the pre-treatments have a positive or negative effect, the authors should use a larger cohort with longer follow-up times.

Response: The conclusion is based on the summary of findings, with unchanged response rates and no obvious dramatic negative effect in local progression, even with short follow-up times. We agree that larger cohorts with longer follow-up is better, but that we do not have at the moment.

  1. What is the point of the 10 years' survival statistics for all the patients (line 144) when most of the patients received the treatments less than 10 years ago? What is the point of the MSS statistical test between the pre-ICI era and the ICI era (p<0.05, line 145) when some unknown fraction of one of the groups receive the treatment 12 or 24 months before the statistical analysis?

Response: For MSS we report 1-year and 3-year survival data. The main reason for figure 1a-c including statistical tests is for transparency, to basically show the data as it is for this quite unique group of patients.

  1. Line 156, p<0.05 - It is not clear how the differences between 93% and 91% showed statistical significance?   

Response: You are completely correct, thanks for the good spotting!!! The p-value for Wieberdink class I-III toxicity should be 0.27 and not 0.05. We have now recalculated all statistics and verified the previous calculations.

  1. Additional minor comments:
  2. Table 1: At the PDF version of the paper mismatched parenthesis.
  3. Lack of title and details on figure legend e.g., p-value, n, etc.
  4. The abbreviation ILP is defined 3 times: lines 13, 21, 49.
  5. The abbreviation ORR is defined twice: lines 30, 52.
  6. The abbreviation CR is defined twice: lines 30, 53.
  7. The abbreviation ICI is first defined at line 44, but mentioned earlier on the text e.g., at line 35.
  8. The abbreviations PFS and OS at line 206 are not defined at the text at all.
  9. Line 115-116 and line 167: ORR= 84% CR=52% while in the table ORR=80% and CR=50% for all patients.
  10. At line 120- “respectively” should be added.
  11. “Number at risk” tables should be added to the Kaplan-Meier graphs. 

Response: All ten remarks have been corrected in the manuscript.

Reviewer 3 Report

Holmberg and Colleagues presented a interesting single center series of melanoma patients treated with ILP and analysed the influence of ICI on patient's selection and outcomes. 

The manuscript is well made and their analysis tried to shed light on a poor investigated issue in the melanoma therapeutic scenario.They also well describe the strengths and weaknesses of their studio.

I have only two observations:

1) the authors could specify a) the type of ICI used (anti PD1, anti CTLA4 or combination); b) the time that elapses between the end of the treatment with ICI and the execution of the PIL

2) Could the authors better describe the therapy performed by patients who had a local or systemic progression in the era of ICI? I know there are few patients who progressed among the 85 treated in the ICI era, but I think it might be very interesting to know the outcomes of patients treated with ICI after ILP.

Author Response

Reviewer 3: Holmberg and Colleagues presented an interesting single center series of melanoma patients treated with ILP and analyzed the influence of ICI on patient's selection and outcomes. 

The manuscript is well made and their analysis tried to shed light on a poor investigated issue in the melanoma therapeutic scenario. They also well describe the strengths and weaknesses of their studio.

I have only two observations:

  1. The authors could specify a) the type of ICI used (anti PD1, anti CTLA4 or combination)

Response: We have now added this information in the first paragraph of the Methods section.

  1. The time that elapses between the end of the treatment with ICI and the execution of the ILP.

Response: Unfortunately, we do not have that specific information available for the patients.

  1. Could the authors better describe the therapy performed by patients who had a local or systemic progression in the era of ICI? I know there are few patients who progressed among the 85 treated in the ICI era, but I think it might be very interesting to know the outcomes of patients treated with ICI after ILP.

Response: Unfortunately, we do not have that specific information available for the patients.

Reviewer 4 Report

This study shows data from a single national reference center for locoregional treatment and compares populations from two different eras: pre-ICI and ICI-era. The study is pertinent and is clear, concise and robust in its results. To improve the study, here are some recommendations:

1) It is not specified what type of follow-up was carried out in both groups, if it was the same or different or if it consisted of the same diagnostic tests.

2) Incomprehensibly, the difference in rates of simultaneous lymph node resection is 12%, however it does not reach statistical significance. It would be good to talk about the reasons for these differences and why appropriate statistical parameters were used to compare. I recommend a new statistical review, exclusively of the two subgroups (pre-ICI vs ICI era).

3) Table 1 should improve the aesthetic aspects, since the results are confused and overlap (for example, "Time from primary tumor to ILP")

4) It would be interesting and more explicit to include the number of patients at risk in the survival graphs (K-M).

5) I miss a graph with the types of ICI used, both before and after, as well as other systemic or local treatments.

6) Some references could be updated or completed with the latest long-term follow-up results with ICI.

Author Response

Reviewer 4: This study shows data from a single national reference center for locoregional treatment and compares populations from two different eras: pre-ICI and ICI-era. The study is pertinent and is clear, concise and robust in its results. To improve the study, here are some recommendations:

  1. It is not specified what type of follow-up was carried out in both groups, if it was the same or different or if it consisted of the same diagnostic tests.

Response: Patients were followed according to Swedish national guidelines, with ultrasound and CT depending on stage. The follow-up regimen has not changed during the study. This has now been clarified in the 3rd paragraph in the Methods.

  1. Incomprehensibly, the difference in rates of simultaneous lymph node resection is 12%, however it does not reach statistical significance. It would be good to talk about the reasons for these differences and why appropriate statistical parameters were used to compare. I recommend a new statistical review, exclusively of the two subgroups (pre-ICI vs ICI era).

Response: You are completely correct, thanks for the good spotting!!! The p-value for lymph node dissections should be 0.06 and not 0.11. The error stemmed from a mislabeling of a variable in the statistics software. We have now recalculated all statistics and verified the previous calculations.

  1. Table 1 should improve the aesthetic aspects, since the results are confused and overlap (for example, "Time from primary tumor to ILP").

Response: This has now been updated.

  1. It would be interesting and more explicit to include the number of patients at risk in the survival graphs (K-M).

Response: Numbers at risk have been added to all K-M graphs.

  1. I miss a graph with the types of ICI used, both before and after, as well as other systemic or local treatments.

Response: Unfortunately, specifics on type of immunotherapies received was not collected in our prospective database.

  1. Some references could be updated or completed with the latest long-term follow-up results with ICI.

Response: We have now updated the reference concerning systemic treatments.